# SAFEGUARDED LEARNED CONVEX OPTIMIZATION WITH GUARANTEED CONVERGENCE

## ABSTRACT

Many applications require quickly and repeatedly solving a certain type of optimization problem, each time with new (but similar) data. However, state-of-the-art general-purpose optimization methods may converge too slowly for real-time use. This shortcoming is addressed by learning to optimize (L2O) schemes, which construct neural networks from parameterized forms of the update operations of general-purpose methods. Inferences by each network form solution estimates, and networks are trained to optimize these estimates for a particular distribution of data. This results in task-specific algorithms (e.g., LISTA, ALISTA, and D-LADMM) that can converge order(s) of magnitude faster than general-purpose counterparts. We provide the first general L2O convergence theory by wrapping all L2O schemes for convex optimization within a single framework. Existing L2O schemes form special cases, and we give a practical guide for applying our L2O framework to other problems. Using safeguarding, our theory proves, as the number of network layers increases, the distance between inferences and the solution set goes to zero, i.e., each cluster point is a solution. Our numerical examples demonstrate the efficacy of our approach for both existing and new L2O methods.

## 1 INTRODUCTION

Solving scientific computing problems often requires application of efficient and scalable optimization algorithms. Despite the ever improving rates of convergence of state-of-the-art general purpose algorithms, their ability to apply to real-time applications is still limited due to the relatively large number of iterations that must be computed. To circumvent this shortcoming, a growing number of researchers use machine learning to develop task-specific algorithms from general-purpose algorithms. For example, inspired by the iterative shrinkage thresholding algorithm (ISTA) for solving the LASSO problem, a sparse coding problem, Gregor & LeCun (2010) proposed to learn the weights in the matrices of the ISTA updates that worked best for a given data set, rather than leave these parameters fixed. They then truncated the method to $K$ iterations, making their Learned ISTA (LISTA) algorithm form a $K$-layer feed-forward neural network. Empirically, their examples showed roughly a 20-fold reduction in computational cost compared to the traditional algorithms. Several related works followed, also demonstrating numerical success (discussed below). While classic optimization results often provide worst-case convergence rates, limited theory exists pertaining to such instances of data drawn from a common distribution (e.g., data supported on a low-dimensional manifold). As a step toward providing such theory, this work addresses the question:

> *Does there exist a universal method that encompasses all L2O algorithms and generates iterates that approach the solution set with guarantees?*

We provide an affirmative answer to this question by prescribing and proving properties of neural networks generated within our L2O framework. Convergence is established by including any choice among several practical safeguarding procedures, including nonmonotone options. Nonmonotone safeguarding enables sequences to traverse portions of the underlying space where the objective function value may increase for a few successive iterations as long as, on average, the sequence approaches the solution set. Although counterintuitive, this ability may lead to faster convergence. Furthermore, we provide a practical guide in our discussion for how practitioners may use our framework to create and apply L2O schemes to their own problems.

The theoretical portion of this work is presented in the context of fixed point theory. This is done to be sufficiently general and provide the desired convergence result for the wide class of optimization methods that can be expressed as special cases of the Krasnosel'skiĭ-Mann (KM) method. For concreteness and ease of application, we then provide the special-case results to several well-known methods (e.g., proximal-gradient, Douglas-Rachford splitting, and ADMM).

**Related Works.**

Learning to learn methods date back decades (e.g., see (Thrun & Pratt, 1998) for a survey of earlier works and references). A seminal L2O work in the context of sparse coding was by Gregor & LeCun (2010). Numerous follow-up papers also demonstrated empirical success at constructing rapid regressors approximating iterative sparse solvers, compression, $\ell_0$ encoding, combining sparse coding with clustering models, nonnegative matrix factorization, compressive sensing MRI, and other applications (Sprechmann et al., 2015; Wang et al., 2016a;b;c;d; Hershey et al., 2014; Yang et al., 2016). A nice summary of unfolded optimization procedures for sparse recovery is given by Ablin et al. (2019) in Table A.1. However, the majority of L2O works pertain to sparse coding and provide limited theoretical results. Some works have interpreted LISTA in various ways to provide proofs of different convergence properties (Giryes et al., 2018; Moreau & Bruna, 2017). Others have investigated structures related to LISTA (Xin et al., 2016; Blumensath & Davies, 2009; Borgerding et al., 2017; Metzler et al., 2017), providing varying results dependent upon the assumptions made. Chen et al. (2018) introduced necessary conditions for the LISTA weight structure to asymptotically achieve a linear convergence rate. This was followed by Liu et al. (2019a), which proved linear convergence of their ALISTA method for the LASSO problem and provided a result stating that, with high probability, the convergence rate of LISTA is at most linear. The mentioned results are useful, yet can require intricate assumptions and proofs specific to the relevant sparse coding problems.

L2O works have also taken other approaches. For example, the paper by Li & Malik (2016) used reinforcement learning with an objective function $f$ and a stochastic policy $\pi^*$ that encodes the updates, which takes existing optimization algorithms as special cases. Our work is related to theirs (cf. Method 1 below and Algorithm 1 in that paper), with the distinction that we include safeguarding and work in the fixed point setting. The idea of Andrychowicz et al. (2016) is to use long short term memory (LSTM) units in recurrent neural networks (RNNs). Additional learning approaches have been applied in the discrete setting (Dai et al., 2018; Li et al., 2018; Bengio et al., 2018). Balcan et al. (2019) reveal how many samples are needed for the average algorithm performance on the training set to generalize over the entire distribution. This is practical for choosing training data and may be used for training L2O networks within our framework.

**Our Contribution.** This is the first work to merge ideas from machine learning, safeguarded optimization, and fixed point theory into a general framework for incorporating data-driven updates into iterative convex optimization algorithms. In particular, given a collection of data and an update operator from an established method (e.g., ADMM or proximal gradient) for solving an optimization problem, we present procedures for creating a neural network that can be used to quickly infer solution estimates. The first novelty of this framework is the ability to incorporate several safeguarding procedures in a general setting. The second is that we present a procedure for utilizing machine learning methods to incorporate knowledge from particular data sets. However, our most significant contribution to the L2O literature is to combine these results into a single, general framework for use by practitioners on any convex optimization problem.

**Outline.** We first provide a brief overview of the fixed point setting of this work in Section 2. Then we present the SKM method and convergence results in Section 3. The incorporation of the SKM method into a neural network and subsequent training approach is presented in Section 4. This is followed in Section 5 by numerical examples, discussion in Section 6, and conclusions in Section 7.

## 2 FIXED POINT METHODS

Let $\mathcal{H}$ be a finite dimensional Hilbert space (e.g., the Euclidean space $\mathbb{R}^n$) with inner product $\langle \cdot, \cdot \rangle$ and norm $\| \cdot \|$. Denote the set of fixed points of each operator $T : \mathcal{H} \to \mathcal{H}$ by $\text{Fix}(T) := \{x \in \mathcal{H} : Tx = x\}$. In this work, for an operator $T$ with a nonempty fixed point set (i.e., $\text{Fix}(T) \neq \emptyset$), the primary problem considered is the fixed point problem:

$$\text{Find } x^\star \in \text{Fix}(T). \tag{1}$$

Convex minimization problems, both constrained and unconstrained, may be equivalently rewritten as the problem (1) for an appropriate mapping $T$. The method chosen for solving the minimization problem determines the operator $T$ in (1) (e.g., see Table 1 below for examples). We focus on the fixed point formulation to provide a general approach, given $T$, for creating a sequence that converges to a solution of (1) and, thus, also of the corresponding optimization problem.

The following definitions will be used in the sequel. A mapping $T : \mathcal{H} \to \mathcal{H}$ is *nonexpansive* if

$$\|Tx - Ty\| \leq \|x - y\|, \quad \text{for all } x, y \in \mathcal{H}. \tag{2}$$

An operator $T : \mathcal{H} \to \mathcal{H}$ is *$\alpha$-averaged* if $\alpha \in (0, 1)$ and there is a nonexpansive operator $Q : \mathcal{H} \to \mathcal{H}$ such that $T = (1 - \alpha)\mathrm{Id} + \alpha Q$, where Id is the identity operator. If the constant $\alpha$ is not important, then $T$ may for brevity be called *averaged*. We also denote the distance between a point $x \in \mathcal{H}$ and a set $C$ by

$$d_C(x) := \inf\{\|x - y\| : y \in C\}. \tag{3}$$

Two operators frequently used in optimization are constructed from monotone relations. Letting $\alpha > 0$ and $f : \mathcal{H} \to \mathbb{R}$ be a function, the *resolvent* of the (possibly) multi-valued subgradient $\partial f$ is

$$J_{\alpha \partial f} := (\mathrm{Id} + \alpha \partial f)^{-1} \tag{4}$$

and the *reflected resolvent* of $\partial f$ is

$$R_{\alpha \partial f} := 2 J_{\alpha \partial f} - \mathrm{Id}. \tag{5}$$

If $f$ is closed, convex, and proper, then the resolvent is precisely the proximal operator, i.e.,

$$J_{\alpha \partial f} = \mathrm{prox}_{\alpha f}(x) := \underset{z \in \mathcal{H}}{\arg \min}\, \alpha f(z) + \frac{1}{2}\|z - x\|^2. \tag{6}$$

From these definitions, it can be shown that $R_{\alpha \partial f}$ is nonexpansive and $J_{\alpha \partial f}$ is averaged. Results above may all be found in Bauschke & Combettes (2017) (e.g., see Prop. 4.4, Thm. 20.25, Example 23.3, and Prop. 23.8). See Table 1 for examples of these operators in optimization methods.

A classic theorem states that sequences generated by successively applying an averaged operator converges to a fixed point. This method comes from Krasnosel'skii (1955) and Mann (1953), which yielded adoption of the name *Krasnosel'skiĭ-Mann* (KM) method. This result is stated below and can be found with various forms and proofs in many works (e.g., see (Bauschke & Combettes, 2017, Thm. 5.14), (Byrne, 2008, Thm. 5.2), (Cegielski, 2012, Thm. 3.5.4), and (Reich, 1979, Thm. 2)).

**Theorem 2.1.** *If an averaged operator $T : \mathcal{H} \to \mathcal{H}$ has a nonempty fixed point set and a sequence $\{x^k\}_{k \in \mathbb{N}}$ with arbitrary $x^1 \in \mathcal{H}$ satisfies the update relation*

$$x^{k+1} = T(x^k), \quad \text{for all } k \in \mathbb{N}, \tag{7}$$

*then there is $x^\star \in \mathrm{Fix}(T)$ such that $\{x^k\}_{k \in \mathbb{N}}$ converges to $x^\star$, i.e., $x^k \to x^\star$.*

There are pathological cases where the result fails for operators that are only nonexpansive (e.g., when $x^1 \neq 0$ and either $T = -\mathrm{Id}$ or $T$ is a rotation). However, this is easily remedied since any convex combination of a nonexpansive operator with the identity is averaged.

## 3 Safeguarded KM Method

This section generalizes the classic KM iteration in (7). We accomplish this by defining an envelope of operators $T_{L2O}(\cdot ; \cdot)$. For a parameter $\zeta$ chosen from an appropriate set, we let $T_{L2O}(\cdot ; \zeta)$ define an operator on $\mathcal{H}$. Changing $\zeta$ may define a new operator with different properties. We do not impose restrictions on $T_{L2O}(\cdot ; \zeta)$ other than it be well-defined, meaning $T_{L2O}(\cdot ; \zeta)$ may fail to be averaged and/or fail to have a fixed point. This is illustrated by the following two examples.

*Example* 3.1. Let $Q : \mathcal{H} \to \mathcal{H}$ be nonexpansive. Then define $T_{L2O} : \mathcal{H} \times \mathbb{R} \to \mathbb{R}$ by

$$T_{L2O}(x; \zeta) := (1 - \zeta)x + \zeta Q(x). \tag{8}$$

For $\zeta \in (0, 1)$, the operator $T_{L2O}(\cdot ; \zeta)$ defined in (8) is $\zeta$-averaged. Although using $\zeta \gg 1$ may result in an operator that fails to be averaged, this can be useful in accelerating the convergence of a method (e.g., see (Giselsson et al., 2016)). $\triangle$

Table 1: Below are the averaged operators for well-known algorithms. We assume $\alpha > 0$ and, when $\alpha$ is multiplied by a gradient, we also assume $\alpha < 2/L$, with $L$ the Lipschitz constant for the gradient. The dual of a function is denoted by a superscript $*$ and $\Omega := \{(x, z) : Ax + Bz = b\}$. The block matrix $M$ is $M = (\alpha^{-1}\text{Id}, A^T; -A, \beta^{-1}\text{Id})$. In each case, $\mathcal{L}$ is the associated Lagrangian.

| Problem | Method | Averaged Operator $T$ |
|---|---|---|
| $\min f(x)$ | Gradient Descent | $\text{Id} - \alpha\nabla f$ |
| $\min f(x)$ | Proximal Point | $\text{prox}_{\alpha f}$ |
| $\min\{g(x) : x \in C\}$ | Projected Gradient | $\text{proj}_C \circ (\text{Id} - \alpha\nabla g)$ |
| $\min f(x) + g(x)$ | Proximal Gradient | $\text{prox}_{\alpha f} \circ (\text{Id} - \alpha\nabla g)$ |
| $\min f(x) + g(x)$ | Peaceman-Rachford | $R_{\alpha\partial f} \circ R_{\alpha\partial g}$ |
| $\min f(x) + g(x)$ | Douglas-Rachford | $\frac{1}{2}\left(\text{Id} + R_{\alpha\partial f} \circ R_{\alpha\partial g}\right)$ |
| $\min_\Omega f(x) + g(z)$ | ADMM | $\frac{1}{2}\left(\text{Id} + R_{\alpha A\partial f^*(A^T\cdot)} \circ R_{\alpha(B\partial g^*(B^T\cdot)-b)}\right)$ |
| $\min f(x)$ s.t. $Ax = b$ | Uzawa | $\text{Id} + \alpha\left(A\nabla f^*(-A^T\cdot) - b\right)$ |
| $\min f(x)$ s.t. $Ax = b$ | Proximal Method of Multipliers | $J_{\alpha\partial\mathcal{L}}$ |
| $\min f(x) + g(Ax)$ | PDHG | $J_{M^{-1}\partial\mathcal{L}}$ |

---

**Method 1** Safeguarded Krasnosel'skiĭ-Mann (SKM)

---
1: Choose $x^1 \in \mathcal{H}$ and $\delta \in (0, 1)$
2: **for** $k = 1, 2, \ldots$ **do**
3:     Choose parameter $\zeta^k$
4:     Choose $\mu_k \in (0, \infty)$
5:     $y^k = T_{L2O}(x^k; \zeta^k)$
6:     **if** $\|S(y^k)\| \leq (1 - \delta)\mu_k$ **then**
7:         $x^{k+1} = y^k$
8:     **else**
9:         $x^{k+1} = T(x^k)$
10:     **end if**
11: **end for**

---

*Example* 3.2. Let $f : \mathcal{H} \to \mathbb{R}$ be closed, convex and proper, and define $T_{L2O} : \mathcal{H} \times (0, \infty) \to \mathbb{R}$ by

$$T_{L2O}(x; \zeta) := \text{prox}_{\zeta f}(x). \tag{9}$$

For fixed $\zeta \in (0, \infty)$, the operator $T_{L2O}(\cdot; \zeta)$ is averaged. $\triangle$

The practicality of $T_{L2O}$ is discussed and demonstrated in Sections 4 and 5, respectively. In the remainder of this work, each operator $T : \mathcal{H} \to \mathcal{H}$ is assumed to be averaged and we set $S :=$ $\text{Id} - T$. Our proposed method below is called the Safeguarded Krasnosel'skiĭ-Mann (SKM) Method. Explanation of the SKM Method is as follows. In Line 1, the initial iterate and parameter $\delta$ are initialized. A common choice for the initial iterate is $x^1 = 0$. The **for** loop on Lines 2 to 11

Table 2: Choices for $\mu_k$ updates that ensure Assumption 3 holds. Here $\alpha \in (0,1)$ and, for fixed $m \in \mathbb{N}$, $\Xi_k$ is the set of the most recent $\min\{m,k\}$ indices for which the inequality in Line 6 holds.

| NAME | UPDATE FORMULA |
|---|---|
| Geometric Sequence
GS($\alpha$) | $$\mu_{k+1} = \begin{cases} (1-\alpha)\mu_k & \text{if Line 6 holds,} \\ \mu_k & \text{otherwise.} \end{cases}$$
*Decrease $\mu_k$ by geometric factor whenever Line 6 holds.* |
| Arithmetic Average

AA | $$m_{k+1} := \begin{cases} m_k + 1 & \text{if Line 6 holds,} \\ m_k & \text{otherwise.} \end{cases}$$
$$\mu_{k+1} := \begin{cases} \dfrac{1}{m_k+1}\left(\|S(x^{k+1})\| + m_k\mu_k\right) & \text{if Line 6 holds,} \\ \mu_k & \text{otherwise.} \end{cases}$$
*Use $m_k$ to count how many times Line 6 holds and $\mu_k$ is the average of the residuals among those times.* |
| Exponential
Moving Average
EMA($\alpha$) | $$\mu_{k+1} := \begin{cases} \alpha\|S(x^{k+1})\| + (1-\alpha)\mu_{k-1} & \text{if Line 6 holds,} \\ \mu_k & \text{otherwise.} \end{cases}$$
*Average $\mu_k$ with the latest residual whenever Line 6 holds.* |
| Recent Term

RT | $$\mu_{k+1} = \begin{cases} \|S(x^k)\| & \text{if Line 6 holds,} \\ \mu_k & \text{otherwise.} \end{cases}$$
*Take $\mu_k$ to be most recent residual for which Line 6 holds.* |
| Recent Max
RM($m$) | $$\mu_{k+1} = \max_{\ell \in \Xi_k} \|S(x^\ell)\|$$
*Take $\mu_k$ to be max of the most recent residuals for which Line 6 holds.* |

generates each update $x^{k+1}$ in the sequence $\{x^k\}_{k\in\mathbb{N}}$. The choice of parameter $\zeta^k$ in Line 3 may be any value that results in a well-defined operator $T_{L2O}(\cdot;\,\zeta^k)$ in Line 5. The choice of $\mu_k$ in Line 4 defines the safeguarding procedure that is used to ensure convergence. Safeguarding is implemented through a descent condition inequality in Line 6. When the inequality in Line 6 holds, $T_{L2O}(x^k;\,\zeta^k)$ is used to update $x^k$ via Line 7. Otherwise, a KM update is used to update $x^k$ via Line 9. Notice also the iteration indexed parameters may all be chosen dynamically (rather than precomputed).

Below are several standard assumptions used to prove our convergence result in Theorem 3.1.

*Assumption* 1. The operator $T$ is nonexpansive with a nonempty fixed point set. ◇

The following assumption ensures boundedness of sequences generated by the SKM method.

*Assumption* 2. The operator $S$ is coercive, i.e.,

$$\lim_{\|x\|\to\infty} \|S(x)\| = \infty. \tag{10}$$

◇

*Remark* 3.1. Assumption 2 does not hold, in general, for nonexpansive operators. For example, if $T$ is the gradient operator $(\mathrm{Id} - \alpha\nabla f)$ for some $\alpha > 0$ and $f$ is a constant function, then $S(x) = 0$ for all $x \in \mathcal{H}$. However, a minor perturbation to the $f$ enables Assumption 2 to hold. In this example, if one fixes small $\varepsilon > 0$ and sets $\tilde{f}(x) := f(x) + \frac{\varepsilon}{2}\|x\|^2$, then the associated $\tilde{S}$ satisfies $\|\tilde{S}(x)\| = \varepsilon\|x\|$. This idea generalizes and, since this works for arbitrarily small $\varepsilon$, in practice it may be reasonable to assume Assumption 2 holds when applying the SKM Method.

---

**Algorithm 2** Learned SKM (LSKM)

---

1: **Stage 1: Initialization/Training.**
2: Choose envelope $T_{L2O}(\cdot; \cdot)$ and network structure $C$, parameterized by $\Theta = (\zeta^k)_{k=1}^K$
3: Choose training loss function $\phi_d$
4: Choose 'optimal' parameter

$$\Theta^\star \in \arg\min_{\Theta \in C} \mathbb{E}_{d \sim \mathcal{D}}\Big[\phi_d(x^K)\Big],$$

   assuming $\mu_k = \infty$ at each layer $k$
5: Choose $\delta$ and safeguarding scheme for $\{\mu_k\}_{k=1}^K$
6: Define the neural network $\mathcal{M} = \mathcal{M}_{\Theta^\star, \delta, \mu_k}$.

7: **Stage 2: Inference.**
8: For input $d$ return $x = \mathcal{M}(d)$

---

*Assumption* 3. If the inequality in Line 6 is satisfied infinitely many times, then the sequence $\{\mu_k\}_{k \in \mathbb{N}}$ converges to zero. ◇

*Remark* 3.2. Assumption 3 may be enforced by using various choices that are dependent upon combinations of the previous residuals. This is illustrated by Table 2 and Corollary 3.1 below.

Our main convergence result is Theorem 3.1 below (proven in the Appendix).

**Theorem 3.1.** *If $\{x^k\}_{k \in \mathbb{N}}$ is a sequence generated by the SKM method and Assumptions 1 to 3 hold, then*

$$\lim_{k \to \infty} d_{\text{Fix}(T)}\left(x^k\right) = 0. \tag{11}$$

*And, if $\{x^k\}_{k \in \mathbb{N}}$ contains a single cluster point, then $\{x^k\}_{k \in \mathbb{N}}$ converges to a point $x^\star \in \text{Fix}(T)$.*

We propose several methods for choosing the sequence $\{\mu_k\}_{k \in \mathbb{N}}$ in Table 2. These methods are *adaptive* in the sense that each update to an iterate $\mu_k$ depends upon the current iterate $x^k$ and (possibly) previous iterates. These update schemes enable each $\mu_k$ to trail the value of the residual norm $\|S(x^k)\|$. This implies there may exist $j \in \mathbb{N}$ for which

$$\|S(x^j)\| < \|S(x^{j+1})\| = \|S\left(T_{L2O}(x^j; \zeta^j)\right)\| \le (1 - \delta)\mu_j. \tag{12}$$

Such leniency is desirable since it is possible that, even though the residual norms converge to zero, constructing a sequence of iterates along the quickest route to the solution set requires traversing portions of the underlying space where the residual norms increase for a few successive iterations. The safeguarding schemes in Table 2 are justified by the Corollary below (proven in the Appendix).

**Corollary 3.1.** *If $\{x^k\}_{k \in \mathbb{N}}$ is a sequence generated by the SKM method and Assumptions 1 and 2 hold and $\{\mu_k\}_{k \in \mathbb{N}}$ is generated using a scheme outlined in Table 2, then Assumption 3 holds and, by Theorem 3.1, the limit (33) holds.*

## 4 A NEURAL NETWORK VIEW

The SKM method may be truncated and executed inferences with a neural network. The input into the network is the data $d$, often in vector form. Each layer of the network consists of an iterate $x^k$, with the output from the final layer $K$ providing the inference estimate $x^K$ that approximates a solution to (1). The feedforward operations between layers are formed by the operator $T_{L2O}(\cdot, \zeta^k)$ if the descent condition in Line 6 of the SKM method holds and $T$ otherwise. We encode all the network parameters with $\Theta := (\zeta^k)_{k=1}^K$. The set $C$, over which $\Theta$ is minimized, may be chosen with great flexibility, the only restriction being that $C$ is a subset of the $K$-tuple set where each $\zeta^k$ is such that $T_{L2O}(\cdot; \zeta^k)$ is well-defined. Since $C$ determines the form of possible $\Theta$, we refer to it in Algorithm 2 as the network structure. For each application of the algorithm, the operators $T$ and $S$ change, depending upon the particular data $d$. For example, the data $d$ could correspond to the measurement vector $d = Ax$ when attempting to recover $x$. Thus, to make explicit this dependence of each operator on the data $d$, we henceforth incorporate a subscript to write $T_d$ and $S_d$.

The "optimal" choice of parameters $\Theta$ depends upon the particular application. Suppose each $d$ is drawn from a common distribution $\mathcal{D}$. Then a choice of "optimal" parameters $\Theta^\star$ may be identified as those for which the expected value of $\phi_d(x^K)$ is minimized, where $\phi_d : \mathcal{H} \to \mathbb{R}$ is an appropriate cost function. In mathematical terms, this is expressed by stating $\Theta^\star$ forms a solution to the problem

$$\min_{\Theta} \mathbb{E}_{d \sim \mathcal{D}} \Big[ \phi_d(x^K(\Theta, d)) \Big], \tag{13}$$

where the expectation $\mathbb{E}_{d \sim D}$ is taken over all $d \in D$ and we emphasize the dependence of $x^K$ on $\Theta$ and $d$ by writing $x^K = x^K(\Theta, d)$. In practice, the problem (13) is approximately solved by using a sample set of data $\{d^n\}_{n=1}^N$ taken from the distribution $\mathcal{D}$ and minimizing an empirical loss function. We summarize the procedure for creating, training, and performing inferences with such L2O networks in Algorithm 2. There $\mathcal{M}$ denotes the neural network, which is dependent upon $\Theta$, $\delta$, and the choice of safeguarding used to construct each $\mu_k$ (see Line 6).

*Remark* 4.1. Different learning problems than (13) may be used (e.g., the min-max problem used by adversarial networks (Goodfellow et al., 2014)).

*Remark* 4.2. Two particular choices of the set $C$ are of interest. The first case is when each parameter $\zeta^k$ may be chosen independently, i.e., the network weights vary by layer. This is used in our numerical examples below. The second case is when $C$ is restricted such that each of its $K$ entries are identical, i.e., the parameters across all layers are fixed so that $\zeta^1 = \zeta^1 = \cdots = \zeta^K$. This latter case corresponds to the structure of recurrent neural networks (RNNs).

## 5 NUMERICAL EXAMPLES

This section presents example L2O schemes applied within the LSKM framework.[1] We numerically investigate (i) the convergence rate of LSKM relative to corresponding KM iterations, (ii) the efficacy of safeguarding procedures when inferences are performed on data for which L2O fails intermittently, and (iii) the convergence of LSKM schemes even when the application of $T_{L2O}$ is not justified theoretically. We first use $T_{L2O}$ from ALISTA (Liu et al., 2019a) on the LASSO Problem. Then we apply the differentiable linearized ADMM of Xie et al. (2019) to a similar problem. We also implement a new L2O scheme for solving the nonnegative least squares (NNLS) problem.

In all experiments, we take $\mu_1 = \|S_d(x^1)\|_2$. Implementations of the LSKM Algorithm are abbreviated by the scheme in Table 2, e.g., we write $\text{GS}(\alpha)$ to mean the Geometric Sequence method with parameter $\alpha$ is used to construct $\{\mu_k\}_{k \in \mathbb{N}}$. As in Algorithm 2, training is completed *without* safeguarding. For notational compactness, in each example we let $f_d^\star$ the optimal value of $f_d(x)$ among all possible $x$. Note each objective function is dependent upon the data $d$, which implies it it is only meaningful to compare curves below that are within the same plot (and not appropriate to compare curves in separate plots). We illustrate the performance in plots using an approximation of the expected objective error

$$\mathbb{E}_{d \sim \mathcal{D}} \left[ f_d(x^k) - f_d^\star \right], \tag{14}$$

using 1,000 test samples. The optimal value $f_d^\star$ for each sample $d$ is estimated by running the KM method for 15,000 iterations.

### 5.1 ALISTA FOR LASSO

Here we consider the popular LASSO problem for sparse coding. Consider an unknown sparse vector $x^\star \in \mathbb{R}^n$ and a matrix $A \in \mathbb{R}^{m \times n}$, which is called the *dictionary*. Then assume we have access to noisy linear measurements $d \in \mathbb{R}^m$, where $\varepsilon \in \mathbb{R}^m$ is additive Gaussian white noise and

$$d = Ax^\star + \varepsilon. \tag{15}$$

Even for underdetermined systems, when $x^\star$ is sufficiently sparse and $\tau \in (0, \infty)$ is an appropriately chosen regularization parameter, $x^\star$ can often be recovered faithfully by solving the LASSO problem

$$\min_{x \in \mathbb{R}^n} f_d(x) := \frac{1}{2} \|Ax - d\|_2^2 + \tau \|x\|_1, \tag{16}$$

---

[1]All of the codes for this work can be found on Github here: (link will be added after review process).

where $\| \cdot \|_2$ and $\| \cdot \|_1$ are the $\ell_2$ and $\ell_1$ norms, respectively. A classic method for solving (16) is the iterative shrinkage thresholding algorithm (ISTA) (e.g., see (Daubechies et al., 2004)), a special case of the proximal-gradient method in Table 1. Given $x^1 \in \mathbb{R}^n$, this method iteratively computes

$$x^{k+1} := T(x^k) := \eta_{\tau/L} \left( x^k - \frac{1}{L} A^T (Ax^k - d) \right), \quad \text{for all } k \in \mathbb{N}, \tag{17}$$

where $L = \|A^t A\|_2$ and $\eta_\theta$ is the soft-thresholding function defined by component-wise operations:

$$\eta_\theta(x) := \text{sgn}(x) \cdot \max\{0, |x| - \theta\}. \tag{18}$$

Implementation details for the LASSO problem may be found in the Appendix.

We applied the LSKM Algorithm to the LASSO problem above by using $T$ defined in (17) and the update operation from ALISTA (Liu et al., 2019a). The tunable operator $T_{L2O}$ is parameterized by $\zeta = (\theta, \gamma)$ for positive scalars $\theta$ and $\gamma$ and defined by

$$T_{L2O}(x; \zeta) := \eta_\theta \left( x - \gamma W^T (Ax - d) \right), \tag{19}$$

with $W$ defined in the Appendix. The parameter $\Theta$ used in Algorithm 2 is $\Theta = (\theta^k, \gamma^k)_{k=1}^K$, which consists of $2K$ scalars. Note $T_{L2O}$ may fail to be averaged, depending on the choice of $\zeta = (\theta, \gamma)$.

The primary illustration of the rapid convergence using LSKM-ALISTA relative to the KM counterpart ISTA is in Figure 1a. There each $x^k$ estimating a solution to (16) is computed for data $d$ drawn from the same distribution $\mathcal{D}_s$ that was used to train the LSKM network. Figure 1b shows a plot comparing performance of LSKM and KM with each $d$ there drawn from a distribution $\mathcal{D}_u$ that is *different* than $\mathcal{D}_s$. For this reason, we refer to $\mathcal{D}_u$ as the *unseen* distribution. For $d \in \mathcal{D}_u$, the L2O scheme ALISTA usually fails to converge without safeguarding whereas the safeguarded method LSKM-ALISTA still converges and much faster than ISTA. The dotted plot with square markers shows the percentage of safeguard activations that occurred in texting. The only two layers in Figure 1b where safeguarding kicked in were layers 2, 7 and 13, with respective percentages 36.7%, 99.6% and 99.2% among 1000 testing samples. Additionally summary results are provided in Table 3. These show the relative usefulness of difference choices of training loss function $\phi_d$, safeguarding method, and performance measured by different test loss functions. In particular, using $\phi_d = f_d$ and EMA(0.25) safeguarding yielded the lowest function value $f_d$; however, instead using GS(0.1) safeguarding resulted in estimates closer to the underlying sparse signals $x^\star$. In both the seen and unseen cases, 20 iterations of LSKM-ALISTA yields better function values $f_d$ than thousands of iterations of ISTA, which reveals orders of magnitude speedup by the safeguarded L2O schemes.

## 5.2 LINEARIZED ADMM

Let $A \in \mathbb{R}^{m \times n}$, $x^\star \in \mathbb{R}^n$, and $d \in \mathbb{R}^m$ be as in Subsection 5.1. Here we apply the L2O scheme differentiatable linearized ADMM (D-LADMM) of Xie et al. (2019) to the closely related sparse coding problem

$$\min_{x \in \mathbb{R}^n} \|Ax - d\|_1 + \tau \|x\|_1. \tag{20}$$

The operators defining $T$ and $T_{L2O}$ for LSKM-D-LADMM are provided in the Appendix along with further implementation details. Comparison plots are provided in Figure 2 and Table 4 summarizes further results.

## 5.3 PROJECTED GRADIENT FOR NONNEGATIVE LEAST SQUARES

Let $A \in \mathbb{R}^{m \times n}$ and $d \in \mathbb{R}^m$. Here we consider the nonnegative least squares (NNLS) problem

$$\min_{x \in \mathbb{R}^n} f_d(x) := \frac{1}{2} \|Ax - d\|_2^2 \quad \text{s.t.} \quad x \geq 0. \tag{21}$$

We proceed by using the projected gradient method in Table 1, where $C := \{x \in \mathbb{R}^n : x \geq 0\}$, $\nabla f_d(x) = A^T (Ax - d)$, and $\text{proj}_C(x) = \max(x, 0)$, with the max applied component-wise. Then, for $\alpha = 1/\|A^T A\|_2$, we take

$$T(x) := \max \left( x - \alpha A^T (Ax - d), 0 \right). \tag{22}$$

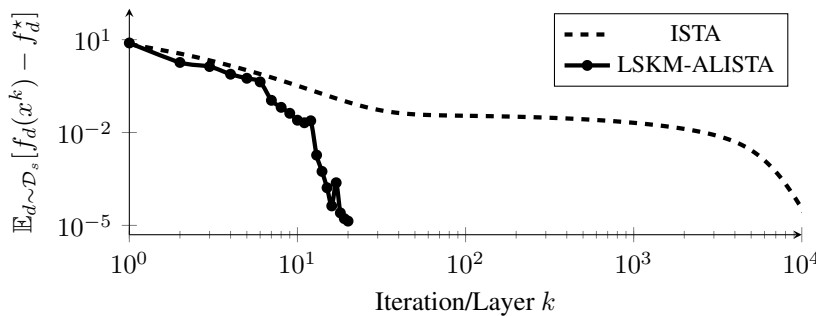

(a) Performance on seen distribution, i.e., $d \sim \mathcal{D}_s$

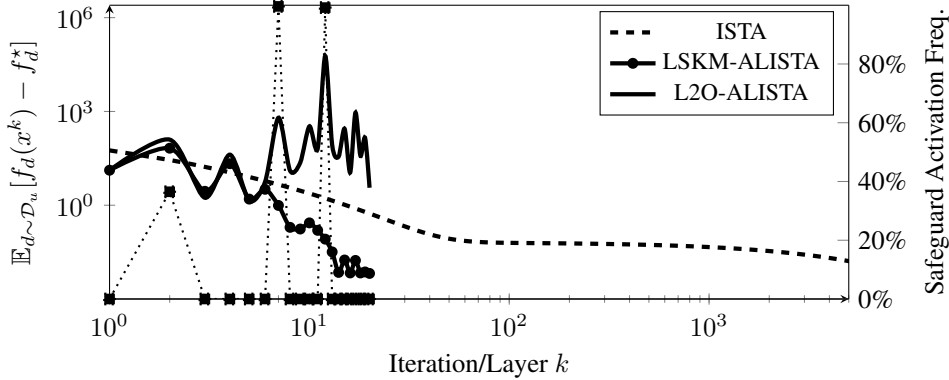

(b) Performance on *unseen* distribution, i.e., $d \sim \mathcal{D}_u$

Figure 1: Expected function value error versus iteration when applied to two different data distributions. Training used $\phi_d = f_d$. Inferences used $\delta = 0.01$ and EMA(0.1).

We then relax $T$ to obtain, for $\zeta = (\alpha, \beta, W)$ with $\alpha \in \mathbb{R}$, $W \in \mathbb{R}^{n \times n}$, and $\beta \in \mathbb{R}^n$, to obtain

$$T_{L2O}(x; \zeta) := \max\left(x - \alpha W^T(Ax - d), \beta\right). \tag{23}$$

In the special case that $W = A^T$, $\alpha = 1/\|A^T A\|_2$, and $\beta = 0$, we recover (22). When safeguarding does not kick in, (23) yields a familiar network structure with the max as the activation function. Here $\Theta = (\zeta^k)_{k=1}^K = (W^k, D^k, \beta^k)_{k=1}^K$ consists of $(n^2 + 2n)K$ trainable parameters. During training we force $\beta^K \geq 0$ to ensure $x^K \in C$. However, note $x^k$ might not be in $C$ for $k < K$. In this example, each learned $W^k$ is likely an approximation of the pseudo-inverse of $A$. Summary plots are given in Figure 3 and more results in Table 5 of the Appendix.

## 6    DISCUSSION

Our numerical examples provide several insights. Tables 3 to 5 in the Appendix reveal the choice of training loss function $\phi_d$ results in different models and performance, even though the minimizers of each $\phi_d$ are identical. This is due to the difference in gradient directions of each $\phi_d$ and reveals the 'best' choice of $\phi_d$ may depend on the application. Figure 1b shows the necessity of safeguarding to ensure convergence and that inferences on unseen data can still be much more rapid than traditional KM methods. For the space limit, we do not focus on timed results. We simply note that the relative cost per iteration of the SKM method versus KM is roughly double in the worst case (since two tentative update operations are needed). However, in cases like our ALISTA example, this relative cost is greatly reduced because some computations (e.g., $Ax^k - d$) are common to both $T_{L2O}$ and $T$. In each example, L2O schemes obtain an order of magnitude reduction in computational cost.

The results of this work reveal a practical procedure for creating and implementing L2O schemes on new problems. Suppose one is presented a convex optimization problem and a standard general purpose method for solving it. The method determines an averaged operator $T$ (e.g., as in Table

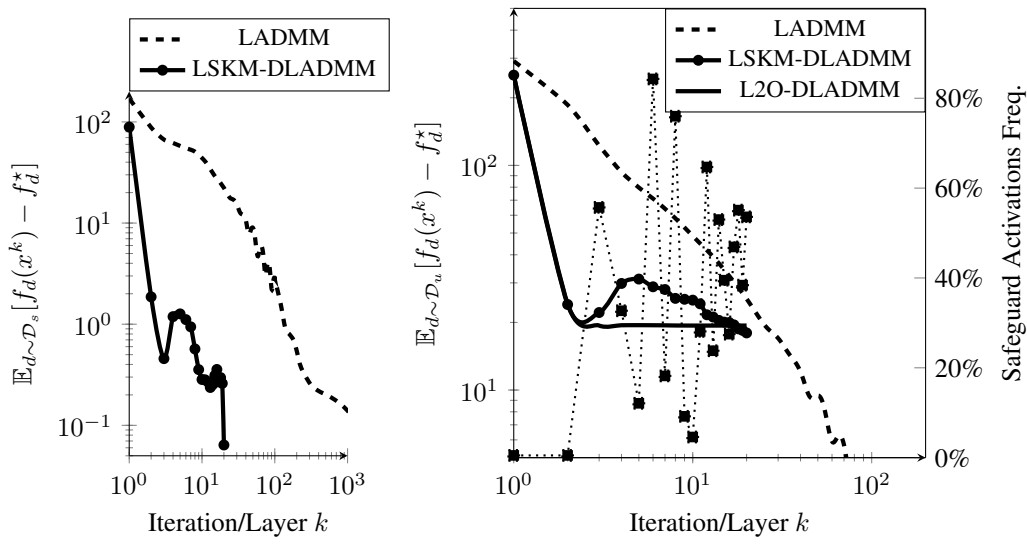

(a) Performance on seen distribution $\mathcal{D}_s$     (b) Performance on *unseen* distribution, i.e., $d \sim \mathcal{D}_u$

Figure 2: Expected relative function value error versus iteration when applied to two different distributions. Training used $\phi_d = f_d$. Inferences used $\delta = 0.01$, and GS(0.1) in (a) and EMA(0.75) in (b).

1). To construct $T_{L2O}$, one may let each scalar, vector, and matrix that does not represent the data $d$ be parameterized (i.e., all of its entries learnable). We may further generalize the operations by replacing scalar parameters with vectors and using element-wise products as in D-LADMM (see (31)). Having $T$ and $T_{L2O}$, a network can be constructed via the LSKM algorithm. For training the network, the learnable parameters may be initialized to the quantities that reduce $T_{L2O}$ to the original operator $T$. Additional structures of parameters may be invoked (e.g., as done by ALISTA with $W$ in (26)). The number of terms to parameterize and how to structure parameterizations are matters subject to the a practitioner's needs/priorities for a given application.

## 7 CONCLUSION

This work establishes a framework for extending the L2O methodology to a wide class of iterative optimization procedures (i.e., KM methods). We provide theoretical results that demonstrate sequences generated by our SKM method possess the property that the distance between each iterate and fixed point set converges to zero. When there is a unique cluster point, this is equivalent to convergence of the method to a solution. Furthermore, our LSKM algorithm provides a straightforward neural network design that inherits the theoretical properties from the SKM method. Practical guidance is also given for constructing learnable operations for new problems. Our numerical experiments demonstrate order(s) of magnitude faster convergence by LSKM implementations than general-purpose counterparts and the efficacy of safeguarding in cases where L2O schemes would otherwise fail to converge. Future work will provide a more efficacious fall-back method for using data-driven updates than classic KM updates and will investigate stochastic extensions with quasi-Féjer monotone operators.

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

## A  APPENDIX

This Appendix contains a subsection with supplemental materials and a subsection with proofs of the theoretical results.

Table 3: Summary of LASSO Problem Results on data $d$ drawn from the seen distribution $\mathcal{D}_s$. $K = 10,000$ iterations of ISTA are used by the KM reference method 'ISTA 10K' while $K = 20$ and $\delta = 0.01$ for the LSKM schemes. 'No guard' refers to LSKM *without* safeguarding.

| Method | ISTA 10K | No guard | EMA(0.1) | EMA(0.1) | EMA(0.1) | GS(0.1) | RM(3) |
|---|---|---|---|---|---|---|---|
| Loss $\phi_d$ | NA | $f_d$ | $\frac{1}{2}\|S_d(\cdot)\|_2^2$ | $\|S_1(\cdot)\|_1$ | $f_d$ | $f_d$ | $f_d$ |
| $R_{f,\mathcal{D}_s}(x^K)$ | 7.72e-04 | 3.36e-04 | 4.65e-04 | 4.58e+00 | 3.33e-04 | 3.32e-04 | 6.68e-04 |
| $\mathbb{E}[\frac{1}{2}\|S(x^K)\|_2^2]$ | 2.11e-09 | 4.68e-07 | 3.75e-07 | 1.58e-05 | 4.68e-07 | 4.68e-07 | 4.95e-07 |

## A.1 NUMERICAL EXAMPLE SUPPLEMENT MATERIALS

**Supplement to Subsection 5.1.** In similar manner to (Chen et al., 2018) and (Liu et al., 2019a), we use the following set up. We take $m = 250$, $n = 500$, and $\tau = 0.001$. Each entry of the dictionary $A$ is sampled i.i.d from the standard Gaussian distribution, i.e., $a_{ij} \sim \mathcal{N}(0, 1/m)$. Having these entries, we then normalize each column of $A$, with respect to the Euclidean norm. Each $d$ in the distribution $\mathcal{D}_s$ of data used to train the neural network is constructed using (15) with noise $\varepsilon \sim 0.1 \cdot \mathcal{N}(0, 1/m)$ and each entry of $x^\star$ as the composition of Bernoulli and Gaussian distributions, i.e., $x_j^\star \sim \text{Ber}(0.1) \circ \mathcal{N}(0, 1)$ for all $j \in [n]$. Each $d$ in the *unseen* distribution $\mathcal{D}_u$ is computed using the same distribution of noise $\varepsilon$ as before and using $x_j^\star \sim \text{Ber}(0.2) \circ \mathcal{N}(1, 2)$. Our data set consists of 10,000 training samples and 1,000 test samples.

If $\tau$ is chosen too large, then the solution to the problem (16) is simply the zero vector, which is undesirable. If $\tau$ is chosen too small, then the solution will not be sparse. Thus, appropriate choice of $\tau$ is crucial. Since the goal of the LASSO problem is to faithfully recover the underlying sparse vector $x^\star$ used to create the measurement $d$, we find it reasonable to choose $\tau$ as a solution estimate of the problem

$$\min_{\tau \in (0,\infty)} \mathbb{E}_{d \sim \mathcal{D}}\left[\|x_d^K - x_d^\star\|_2^2\right], \tag{24}$$

where $x_d^K$ is the output of ISTA after $K = 15,000$ iterations for the problem (16) and $x_d^\star$ is the sparse vector from which $d$ was created. To simplify (24), we assume $\tau$ was of the form $t = 10^\alpha$ with integer $\alpha \in \mathbb{Z}$ to obtain the problem

$$\min_{\alpha \in \mathbb{Z}} \mathbb{E}_{d \sim \mathcal{D}}\left[\|x_d^K - x_d^\star\|_2^2\right], \tag{25}$$

Approximately solving this discrete problem for our data set revealed the optimal choice yielded $\alpha = -3$ and $\tau = 0.001$.

In practice, such a choice of $\tau$ may be difficult to ascertain. However, if the practitioner has access to each underlying $x_d^\star$ for their training data $d$, then one can circumvent the problem of choosing $\tau$ by simply using the training loss function $\phi_d(x) := \|x - x_d^\star\|_2^2$. This was precisely the approach taken in Liu et al. (2019a).

To define the learned operations in $T_{L2O}$, we let

$$W \in \arg\min_{M \in \mathbb{R}^{m \times n}} \|M^T A\|_F, \quad \text{s.t.} (M_{:,\ell})^T A_{:,\ell} = 1, \quad \text{for all } \ell \in [n], \tag{26}$$

where $\|\cdot\|_F$ is the Frobenius norm and the Matlab notation $M_{:,\ell}$ is used to denote the $\ell$th column of the matrix $M$.

**Supplement to Subsection 5.2.** LADMM is used to solve problems of the form

$$\min_{x \in \mathbb{R}^n, z \in \mathbb{R}^m} f(x) + g(z) \quad \text{s.t.} \quad Ax + Bz = d, \tag{27}$$

for which LADMM generates sequences $\{x^k\}_{k \in \mathbb{N}}$, $\{z^k\}_{k \in \mathbb{N}}$ and $\{\nu^k\}_{k \in \mathbb{N}}$ defined by the updates

$$\begin{aligned} x^{k+1} &:= \text{prox}_{\beta f}\left(x^k - \beta A^T\left[\nu^k + \alpha\left(Ax^k + Bz^k - d\right)\right]\right), \\ z^{k+1} &:= \text{prox}_{\gamma g}\left(z^k - \gamma B^T\left[\nu^k + \alpha\left(Ax^{k+1} + Bz^k - d\right)\right]\right), \\ \nu^{k+1} &:= \nu^k + \alpha\left(Ax^{k+1} + Bz^{k+1} - d\right), \end{aligned} \tag{28}$$

Table 4: Summary of D-LADMM Problem Results on data $d$ drawn from the seen distribution $\mathcal{D}_s$. $K = 1,000$ iterations of LADMM are used by the KM reference method 'LADMM 1K' while $K = 20$ and $\delta = 0.01$ for the LSKM schemes. 'No guard' refers to LSKM *without* safeguarding.

| Method | LADMM 1K | No guard | EMA(0.75) | GS(0.1) | RT |
|---|---|---|---|---|---|
| Loss $\phi_d$ | NA | $f_d$ | $f_d$ | $f_d$ | $f_d$ |
| $R_{f,\mathcal{D}_s}(x^K)$ | 3.48e+00 | 1.78e+00 | 1.52e+01 | 1.78e+00 | 1.28e+01 |

with given scalars $\alpha, \beta, \gamma \in (0, \infty)$. The problem (20) may be written in the form of (27) by taking $f = \tau \| \cdot \|_1$, $g = \| \cdot \|_1$, and $B = -\mathrm{Id}$. In this case, the proximal operators in (28) reduce to soft-thresholding operators. Although not given in Table 1, the update $\nu^{k+1}$ is generated by applying an averaged operator $T$ to $\nu^k$. (This follows since LADMM may expressed as a special case of proximal ADMM, which itself is a special case of the ADMM method. The details of this derivation are outside the scope of this paper.) This implies

$$\|S(\nu^k)\| = \|\nu^k - T(\nu^k)\| = \|\nu^k - \nu^{k+1}\| = \alpha\|Ax^{k+1} - z^{k+1} - d\|. \tag{29}$$

Because we compare methods that use differing choices of $\alpha$, the residual comparison illustrations presented below will, for consistency of interpretation, assume $\alpha = 1$ in (29). And, although $x^{k+1}$ and $z^{k+1}$ form intermediate computations, for notational clarity, the term $x^k$ in the SKM and LSKM schemes is replaced in this subsection by the tuple $(x^k, z^k, \nu^k)$. This is of practical importance too since it is the sequence $\{x^k\}_{k\in\mathbb{N}}$ that converges to a solution of (20).

We now modify the iteration (28) for the problem (20) to create the D-LADMM L2O scheme. We generalize soft-thresholding to vectorized soft-thresholding for $\beta \in \mathbb{R}^n$ by

$$\eta_\beta(x) = (\eta_{\beta_1}(x_1),\ \eta_{\beta_2}(x_2),\ \ldots,\ \eta_{\beta_n}(x_n)). \tag{30}$$

We assume $\eta_\beta$ represents the scalar soft-thresholding in (18) when $\beta \in \mathbb{R}$ and the vector generalization (30) when $\beta \in \mathbb{R}^n$. Combining ideas from ALISTA (Liu et al., 2019a) and D-LADMM (Liu et al., 2019b), given $(x^k, z^k, \nu^k) \in \mathbb{R}^n \times \mathbb{R}^m \times \mathbb{R}^m$, $\alpha^k, \gamma^k, \xi^k \in \mathbb{R}^m$, $\beta^k, \sigma^k \in \mathbb{R}^n$, $W_1 \in \mathbb{R}^{n \times m}$, and $W_2 \in \mathbb{R}^{m \times m}$, set

$$\begin{aligned}
\tilde{x}^{k+1} &:= \eta_{\beta^k}\left(x^k - \sigma^k \circ (W_1^k)^T \left[\nu^k + \alpha_k \circ \left(Ax^k - z^k - d\right)\right]\right), \\
\tilde{z}^{k+1} &:= \eta_{\gamma^k}\left(z^k - \xi^k \circ (W_2^k)^T \left[\nu^k + \alpha_k \circ \left(A\tilde{x}^{k+1} - z^k - d\right)\right]\right), \\
\tilde{\nu}^{k+1} &:= \nu^k + \alpha_k \circ \left(A\tilde{x}^{k+1} - \tilde{z}^{k+1} - d\right),
\end{aligned} \tag{31}$$

with element-wise products denoted by $\circ$. For the parameter $\zeta^k := (\alpha^k, \beta^k, \gamma^k, \sigma^k, \xi^k, W_1^k, W_2^k)$, then define

$$T_{L2O}(x^k, z^k, \nu^k;\ \zeta^k) := (\tilde{x}^{k+1}, \tilde{z}^{k+1}, \tilde{\nu}^{k+1}). \tag{32}$$

Fixing the number of iterations $K$, the learnable parameters from (31) used in the LSKM Algorithm may be encoded by $\Theta = (\zeta^k)_{k=1}^K = \left(\alpha^k, \beta^k, \gamma^k, \sigma^k, \xi^k, W_1^k, W_2^k\right)_{k=1}^K$, consisting of $(2n + 3m + mn + m^2)K$ scalars.

For data generation, we follow the same settings as in the experiments of Subsection 5.1. Although the value of $\tau$ is not optimally chosen, this was used as it worked well for D-LADMM.

**Supplement to Subsection 5.3.** We take $m = 500$, $n = 250$, $a_{ij} \sim \mathrm{Ber}(0.1) \cdot \mathrm{rand}[0, 1]$. We use noise $\varepsilon \sim 0.1 \cdot \mathcal{N}(0, 1/m)$. Each $d \sim \mathcal{D}_s$ used for training the neural network is sampled using $d = Ax^\star + \varepsilon$ with $x \sim \max(\mathcal{N}(0, 1), 0)$. For unseen data $d \sim \mathcal{D}_u$ we sample $x^\star \sim \max(\mathcal{N}(5, 10^2), 0)$. We sample 10,000 training samples from $\mathcal{D}_s$ to train the neural network and 1,000 samples from $\mathcal{D}_s$ and $\mathcal{D}_u$, respectively, for testing.

## A.2 PROOFS

For ease of reference, we restate Theorem 3.1 below. Then provide a proof is provided for it.

**Theorem 3.1.** If $\{x^k\}_{k\in\mathbb{N}}$ is a sequence generated by the SKM method and Assumptions 1 to 3 hold, then

$$\lim_{k\to\infty} d_{\mathrm{Fix}(T)}\left(x^k\right) = 0. \tag{33}$$

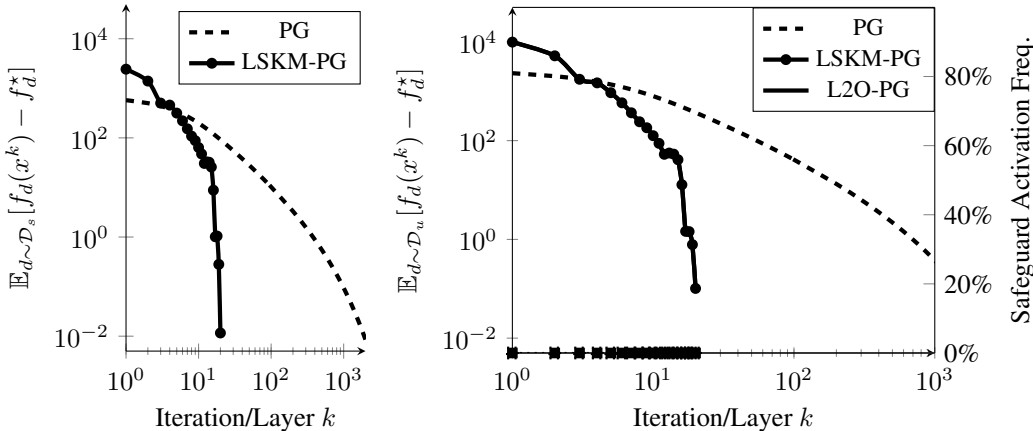

(a) Performance on seen distribution $\mathcal{D}_s$      (b) Performance on *unseen* distribution, i.e., $d \sim \mathcal{D}_u$

Figure 3: Expected function value error versus iteration when applied to two different data distributions. Training used $\phi_d = f_d$. Inferences used $\delta = 0.01$ and GS(0.1).

Table 5: Summary of NNLS Problem Results on data $d$ drawn from the seen distribution $\mathcal{D}_s$. $K = 1,500$ iterations of PG are used by the KM reference method 'PG 1.5K' while $K = 20$ and $\delta = 0.01$ for the LSKM schemes. 'No guard' refers to LSKM *without* safeguarding.

| Method | PG 1.5K | No guard | EMA(0.75) | GS(0.1) | RT |
|---|---|---|---|---|---|
| Loss $\phi_d$ | NA | $f_d$ | $f_d$ | $f_d$ | $f_d$ |
| $R_{f,\mathcal{D}_s}(x^K)$ | 3.80e+06 | 1.86e+06 | 2.37e+07 | 1.86e+06 | 1.65e+08 |

And, if $\{x^k\}_{k\in\mathbb{N}}$ contains a single cluster point, then $\{x^k\}_{k\in\mathbb{N}}$ converges to a point $x^\star \in \text{Fix}(T)$.

*Proof.* If the inequality in Line 6 holds finitely many times, then there exists an index beyond which Line 9 is always used to update $x^k$. In this case, for large $k$ the SKM Method takes the form of the classic KM Method, which is known to converge (e.g., see (Cegielski, 2012, Theorem 3.7.1), (Groetsch, 1972, Corollary 3), and (Bauschke & Combettes, 2017, Theorem 5.15)). Thus, it is sufficient to only consider the case where the inequality in Line 6 holds infinitely many times. We proceed by first showing the sequence $\{x^k\}_{k\in\mathbb{N}}$ is bounded (Step 1). This is used to prove $\{x^k\}_{k\in\mathbb{N}}$ has a cluster point in $\text{Fix}(T)$ (Step 2). Results from these steps are then applied to obtain the desired limit (33) (Step 3).

**Step 1:** By Assumption 2, there exists $R \in (0, \infty)$ sufficiently large to ensure

$$\|x\| > R \implies \|S(x)\| > 1, \quad \text{for all } x \in \mathcal{H}. \tag{34}$$

Equivalently, we may write

$$\|S(x)\| \leq 1 \implies \|x\| \leq R, \quad \text{for all } x \in \mathcal{H}. \tag{35}$$

By Assumption 3, there also exists $N_1 \in \mathbb{N}$ such that

$$\mu_k \leq 1, \quad \text{for all } k \geq N_1. \tag{36}$$

Fix any $z \in \text{Fix}(T)$. We claim

$$\|x^k - z\| \leq \max_{\ell \in [N_1]} \{2R, \|x^\ell - z\|\}, \quad \text{for all } k \in \mathbb{N}. \tag{37}$$

The result (37) holds trivially for all $k \in [N_1]$. Proceeding by induction, suppose (37) holds for some $k > N_1$. If the inequality in Line 6 holds, then (35), (36) and the update formula in Line 7 together imply $\|x^{k+1}\| \leq R$. Since $\|S(z)\| = 0$, (35) also implies $\|z\| \leq R$. Thus,

$$\|x^{k+1} - z\| \leq \|x^{k+1}\| + \|z\| \leq 2R \leq \max_{\ell \in [N_1]} \{2R, \|x^\ell - z\|\}. \tag{38}$$

If instead the update in Line 9 is applied, the averagedness of $T$ implies there is $\alpha \in (0, 1)$ such that

$$\|x^{k+1} - z\|^2 \le \|x^k - z\|^2 - \frac{1 - \alpha}{\alpha}\|S(x^k)\|^2 \tag{39}$$

(e.g., see Prop. 4.35 in (Bauschke & Combettes, 2017) or Cor. 2.2.15 and Cor. 2.2.17 in (Cegielski, 2012)), and so

$$\|x^{k+1} - z\| \le \|x^k - z\| \le \max_{\ell \in [N_1]}\{2R, \|x^\ell - z\|\}. \tag{40}$$

Equations (38) and (40) together close the induction, from which (37) follows. Whence

$$\|x^k\| \le \|x^k - z\| + \|z\| \le \max_{\ell \in [N_1]}\{2R, \|x^\ell - z\|\} + R, \quad \text{for all } k \in \mathbb{N}, \tag{41}$$

which verifies the sequence $\{x^k\}_{k \in \mathbb{N}}$ is bounded.

**Step 2:** Because the inequality in Line 6 holds infinitely many times, there exists a subsequence $\{x^{q_k}\}_{k \in \mathbb{N}} \subseteq \{x^k\}_{k \in \mathbb{N}}$ satisfying

$$0 \le \lim_{k \to \infty} \|S(x^{q_k})\| \le \lim_{k \to \infty} (1 - \delta)\mu_k = 0, \tag{42}$$

from which the squeeze theorem asserts $\|S(x^{q_k})\| \to 0$. Since $\{x^k\}_{k \in \mathbb{N}}$ is bounded, so also is $\{x^{q_k}\}_{k \in \mathbb{N}}$. Thus, there exists a subsequence $\{x^{\ell_k}\}_{k \in \mathbb{N}} \subseteq \{x^{q_k}\}_{k \in \mathbb{N}}$ converging to a limit $p \in \mathcal{H}$. Then applying the fact $S$ is 2-Lipschitz and $\|\cdot\|$ is continuous yields

$$0 = \lim_{k \to \infty} \|S(x^{\ell_k})\| = \left\|S\left(\lim_{k \to \infty} x^{\ell_k}\right)\right\| = \|S(p)\| \quad \Longrightarrow \quad p \in \text{Fix}(T). \tag{43}$$

That is, $\{x^k\}_{k \in \mathbb{N}}$ contains a cluster point $p \in \text{Fix}(T)$.

**Step 3:** Let $\varepsilon > 0$ be given. Following (Bauschke et al., 2014, Def. 2), define the $\varepsilon$-enlargement

$$\text{Fix}(T)_{[\varepsilon]} := \{x \in \mathcal{H} : d_{\text{Fix}(T)}(x) \le \varepsilon\}. \tag{44}$$

Note $\text{Fix}(T)_{[\varepsilon]}$ is a nonempty closed and bounded subset of $\mathcal{H}$. Set

$$C := \left(B(0, R) - \text{Fix}(T)_{[\varepsilon/2]}\right) \cup \partial\text{Fix}(T)_{[\varepsilon/2]}, \tag{45}$$

where $B(0, R)$ is the closed ball of radius $R$ centered at the origin. Because $\mathcal{H}$ is finite dimensional and $C$ is closed and bounded, $C$ is compact. Thus, every continuous function obtains its infimum over $C$. In particular, we may set

$$\zeta = \min_{x \in C} \|S(x)\|. \tag{46}$$

Note $\zeta > 0$ since $C \cap \text{Fix}(T) = \emptyset$. Consequently, letting $\tilde{\zeta} := \min\{1, \zeta/2\}$ yields

$$\|S(x)\| \le \tilde{\zeta} \quad \Longrightarrow \quad x \in \text{Fix}(T)_{[\varepsilon]} \quad \Longrightarrow \quad d_{\text{Fix}(T)}(x) \le \varepsilon, \quad \text{for all } x \in \mathcal{H}, \tag{47}$$

where the first implication holds because $x \in B(0, R)$ by (35) and $x \notin C$ by (46). By Assumption 3, there exists $N_2 \in \mathbb{N}$ such that

$$\mu_k \le \tilde{\zeta}, \quad \text{for all } k \ge N_2. \tag{48}$$

By the result of Step 2, there exists $N_3 \ge N_2$ such that

$$\|x^{N_3} - p\| \le \varepsilon \quad \Longrightarrow \quad d_{\text{Fix}(T)}(x^{N_3}) \le \varepsilon. \tag{49}$$

We claim

$$d_{\text{Fix}(T)}(x^k) \le \varepsilon, \quad \text{for all } k \ge N_3, \tag{50}$$

which, by the arbitrariness of $\varepsilon$, implies (33) holds. Indeed, inductively suppose (50) holds for some $k \ge N_3$. If the inequality in Line 6 holds, then (47) and (48) together imply

$$\|S(x^{k+1})\| \le (1 - \delta)\mu_k \le \tilde{\zeta} \quad \Longrightarrow \quad d_{\text{Fix}(T)}(x^{k+1}) \le \varepsilon. \tag{51}$$

Otherwise, letting $P : \mathcal{H} \to \mathcal{H}$ be the projection operator onto $\text{Fix}(T)$, we deduce

$$\|x^{k+1} - P(x^{k+1})\| \le \|x^{k+1} - P(x^k)\| \le \|x^k - P(x^k)\| = d_{\text{Fix}(T)}(x^k) \le \varepsilon, \tag{52}$$

where the second inequality follows from (39), taking $z = P(x^k)$. Note the left hand side of (52) equals the distance from $x^{k+1}$ to $\text{Fix}(T)$. Therefore, (51) and (52) close the induction in each case, and (33) holds.

The limit (33) can be used in a similar manner to the work in Step 2 above to prove each cluster point of $\{x^k\}_{k \in \mathbb{N}}$ is in $\text{Fix}(T)$. Thus, if $\{x^k\}_{k \in \mathbb{N}}$ admits a unique cluster point, then the entire sequence converges to a point $x^\star \in \text{Fix}(T)$. $\qquad\square$

We restate Corollary 3.1 below and then provide a proof.

**Corollary 3.1.** If $\{x^k\}_{k \in \mathbb{N}}$ is a sequence generated by the SKM method and Assumptions 1 and 2 hold and $\{\mu_k\}_{k \in \mathbb{N}}$ is generated using a scheme outlined in Table 2, then Assumption 3 holds and, by Theorem 3.1, the limit (33) holds.

*Proof.* The proof is parsed into four parts, one for each particular choice of the sequence $\{\mu_k\}_{k \in \mathbb{N}}$ in Table 2, where we note "Recent Term" is a special case of "Recent Max" obtained by taking $m = 1$. Each proof part is completely independent of the others and is separated by italic text. However, to avoid excessive writing, in each section let $\Gamma \subseteq \mathbb{N}$ be the set of all indices for which the descent condition in Line 6 holds, the sequence $\{t_k\}_{k \in \mathbb{N}}$ be an ascending enumeration of $\Gamma$, $m_k$ be the number of times the descent condition has been satisfied by iteration $k$, and $\mu_1 \in (0, \infty)$.

*Geometric Seqeunce.* Define the sequence $\{\mu_k\}_{k \in \mathbb{N}}$ using, for each $k \in \mathbb{N}$, the update formula

$$\mu_{k+1} = \begin{cases} \mu_k & \text{if Line 6 holds,} \\ (1 - \delta)\mu_k & \text{otherwise.} \end{cases} \tag{53}$$

This implies

$$\mu_k = (1 - \delta)^{m_k} \mu_1. \tag{54}$$

Since $\Gamma$ is infinite, $\lim_{k \to \infty} m_k = \infty$, and it follows that

$$\lim_{k \to \infty} \mu_k = \lim_{k \to \infty} (1 - \delta)^{m_k} \mu_1 = 0 \cdot \mu_1 = 0, \tag{55}$$

i.e., Assumption 3 holds.

*Arithmetic Average.* Define the sequence $\{\mu_k\}_{k \in \mathbb{N}}$ using, for each $k \in \mathbb{N}$, the update formula

$$\mu_{k+1} := \begin{cases} \dfrac{1}{m_k + 1} \left( \|S(x^{k+1})\| + m_k \mu_k \right) & \text{if Line 6 holds,} \\ \mu_k & \text{otherwise.} \end{cases} \tag{56}$$

Then observe

$$0 \le \mu_{t_k+1} \le \frac{(1 - \delta)\mu_{t_k} + m_{t_k} \mu_{t_k}}{m_{t_k} + 1} = \left( 1 - \frac{\delta}{m_{t_k} + 1} \right) \mu_{t_k} \le \mu_{t_k}, \quad \text{for all } k \in \mathbb{N}. \tag{57}$$

Since $\mu_{k+1} = \mu_k$ whenever $k \notin \Gamma$, (57) shows $\{\mu_k\}_{k \in \mathbb{N}}$ is monotonically decreasing. Consequently, using induction reveals

$$0 \le \mu_{t_k} - \frac{\delta}{m_{t_k} + 1} \mu_{t_k} \le \mu_1 - \sum_{\ell=1}^{k} \frac{\delta \mu_{t_\ell}}{m_{t_\ell} + 1} = \mu_1 - \sum_{\ell=1}^{k} \frac{\delta \mu_{t_\ell}}{\ell + 1} \quad \text{for all } k \in \mathbb{N}, \tag{58}$$

where we note $m_{t_\ell} = \ell$ in the sum since $m_\ell$ increments once each time a modification occurs in the sequence $\{\mu_k\}_{k \in \mathbb{N}}$. By way of contradiction, suppose there exists $\alpha \in (0, \infty)$ such that

$$\liminf_{k \to \infty} \mu_k \ge \alpha > 0. \tag{59}$$

Then (58) implies

$$\sum_{\ell=1}^{k} \frac{\alpha \delta}{\ell + 1} \le \sum_{\ell=1}^{k} \frac{\delta \mu_{t_\ell}}{\ell + 1} \le \mu_1, \quad \text{for all } k \in \mathbb{N}. \tag{60}$$

However, the sum on the left hand side becomes a divergent harmonic series as $k \to \infty$, contradicting the finite upper bound on the right hand side. This contradiction proves assumption (59) is false, from which it follows that

$$\liminf_{k \to \infty} \mu_k = 0. \tag{61}$$

By the monotone convergence theorem, we deduce $\mu_k \to 0$, i.e., Assumption 3 holds.

*Exponential Moving Average.* Given $\theta \in (0, 1)$, for all $k \in \mathbb{N}$, define

$$\mu_{k+1} := \begin{cases} \theta \|S(x^{k+1})\| + (1-\theta)\mu_{k-1} & \text{if Line 6 holds,} \\ \mu_k & \text{otherwise.} \end{cases} \tag{62}$$

Now observe

$$\mu_{t_k+1} = \theta\|S(x^{t_k+1})\| + (1-\theta)\mu_{t_k} \le \theta(1-\delta)\mu_{t_k} + (1-\theta)\mu_{t_k} = (1-\theta\delta)\mu_{t_k}. \tag{63}$$

This shows the sequence $\{\mu_k\}_{k \in \mathbb{N}}$ is nonincreasing and, when a decrease does occur, it is by a geometric factor of the current iterate. Through induction, it follows that

$$\mu_k \le (1-\theta\delta)^{m_k}\mu_1, \quad \text{for all } k \in \mathbb{N}. \tag{64}$$

Since $\Gamma$ is infinite, $\lim_{k \to \infty} m_k = \infty$. This, combined with the fact $(1-\theta\delta) \in (0,1)$, implies

$$0 \le \lim_{k \to \infty} \mu_k \le \lim_{k \to \infty} (1-\theta\delta)^{m_k}\mu_1 = 0 \cdot \mu_1 = 0, \tag{65}$$

from which the squeeze theorem asserts Assumption 3 holds.

*Recent Max.* Let $m \in \mathbb{N}$. Set $\Xi_k$ to be the set of the most recent $\min\{m, k\}$ indices for which the descent condition in Line 6 held, where $\{\mu_k\}_{k \in \mathbb{N}}$ is defined, for all $k \in \mathbb{N}$, by the update formula

$$\mu_{k+1} = \begin{cases} \max_{\ell \in \Xi_k} \|S(x^\ell)\| & \text{if Line 6 holds,} \\ \mu_k & \text{otherwise.} \end{cases} \tag{66}$$

The sequence $\{\mu_k\}_{k \in \mathbb{N}}$ is monotonically decreasing since the inequality in Line 6 implies, each time a new term $\|S(x^k)\|$ is introduced so that $\|S(x^k)\| \in \Xi_{k+1}$, the new term is no larger than the largest term in $\Xi_k$. All that remains is to show this sequence converges to zero. By way of contradiction, suppose there exists $\alpha \in (0, \infty)$ such that

$$\liminf_{k \to \infty} \mu_k = \alpha > 0. \tag{67}$$

Then choose

$$\varepsilon = \frac{\delta\alpha}{2(1-\delta)}, \tag{68}$$

which implies

$$(1-\delta)(\alpha + \varepsilon) < \alpha. \tag{69}$$

By (67) and the fact $\Gamma$ is infinite, there exists $\tilde{N} \in \mathbb{N}$ with $\tilde{N} > m$ such that

$$\|\mu_{t_{\tilde{N}}} - \alpha\| < \varepsilon \implies \mu_{t_{\tilde{N}}} < \alpha + \varepsilon. \tag{70}$$

Then note each new element to $\Xi_k$ is no larger than $(1-\delta)\mu_{t_{\tilde{N}}}$. And, for any $k$ after $m$ such replacements occur, it follows that

$$\mu_k = \max_{\ell \in \Xi_k} \|S(x^\ell)\| \le (1-\delta)\mu_{t_{\tilde{N}}} \le (1-\delta)(\alpha + \varepsilon) < \alpha, \tag{71}$$

a contradiction to (67). This contradiction shows our assumption (67) must be false, and so

$$\liminf_{k \to \infty} \mu_k = 0. \tag{72}$$

By the monotone convergence theorem, we conclude Assumption 3 holds. □

