# OpenReview forum: "Universal Safeguarded Learned Convex Optimization with Guaranteed Convergence"
_ICLR.cc/2020/Conference — Reject_

### Official Review · AnonReviewer2 · 2019-10-23
**Official Blind Review #2**

**Rating:** 3

**Review:**

This paper proposes a framework to unfold the safeguarded Krasnosel’ski˘ı-Mann (SKM) method for the learn to optimization (L2O) schemes. First, SKM is proposed in Algorithm 1 with convergence guarantee established in Theorem 3.1 and Corollary 3.1. Then, SKM is unfolded and executed with a neural network summarized in Algorithm 2. Experiments on the Lasso and nonnegative least squares show the efficiency of the proposed method as well as the effectiveness of safeguarding compared to traditional L2O methods.

Advantages:
1. A general framework that encompasses all L2O algorithms for use by practitioners on any convex optimization problem.
2. It seems that the convergence analysis of Krasnosel’ski˘ı-Mann equipped with safegarding is established for the first time.

Weakness:
The idea of reimplementing an iterative algorithm in a deep architecture is not new, and the combination of safegarding with KM has already been analyzed [1,2].  Moreover, the experiments are not convincing.
1. Safegarding is the key point of this paper, but the authors did not review related works on safegarding. Please show the relationships of SKM with prior works and comment on the novelty of the analysis in this paper.
[1] Themelis, Andreas, and Panagiotis Patrinos. "SuperMann: a superlinearly convergent algorithm for finding fixed points of nonexpansive operators." IEEE Transactions on Automatic Control (2019).
[2] Sopasakis, Pantelis, et al. "A primal-dual line search method and applications in image processing." 2017 25th European Signal Processing Conference (EUSIPCO). IEEE, 2017.

2. All the 3 experiments are conducted on synthetic datasets which is not convincing enough to show the efficiency and effectiveness of LSKM. It is suggested to carry out experiments on real-world datasets like [3,4] with state-of-the-art methods.
[3] Sun, Jian, Huibin Li, and Zongben Xu. "Deep ADMM-Net for compressive sensing MRI." Advances in neural information processing systems. 2016.
[4] Metzler, Chris, Ali Mousavi, and Richard Baraniuk. "Learned D-AMP: Principled neural network based compressive image recovery." Advances in Neural Information Processing Systems. 2017.

3. The are too many errors in references, for examples:
(3.1) What is "In S. Bengio, H.Wallach, H. Larochelle, K. Grauman, N. Cesa-Bianchi, and R. Garnett (eds.), Advances in Neural Information Processing Systems 31"? This error appears multiple times.
(3.2) Show complete information of reference "Liu et al. (2019a)".


**Experience Assessment:**

I have read many papers in this area.

**Review Assessment: Checking Correctness Of Derivations And Theory:**

I carefully checked the derivations and theory.

**Review Assessment: Checking Correctness Of Experiments:**

I assessed the sensibility of the experiments.

**Review Assessment: Thoroughness In Paper Reading:**

I read the paper at least twice and used my best judgement in assessing the paper.

---

> ### Author Response · Authors · 2019-11-15
> **Response to Reviewer #2**
>
> Thank you for your constructive feedback and insightful comments concerning the connection between our work and other existing papers. We believe our responses below address your concerns and hope, upon reading these, you will increase your score.
>
> A main point to start with: it is a theory paper, and perhaps the first set of convergence/robustness theories ever offered for the important field of learning-to-optimize. It is NOT “yet another” application-driven work that turns algorithms into deep architectures (which we are also very familiar with).
>
> 1) To be brief:
>
> 1.a) “The idea of reimplementing an iterative algorithm in a deep architecture is not new” - we agree; that is not the point of our paper either.
>
> 1.b) The key point of this paper is NOT just safeguarding; but rather, safeguarding applied to learning-to-optimize for the first time, and a unified convergence theory for (convex) learning-to-optimize, also for the first time. Neither of the mentioned safeguarding papers, nor any other literature we’re aware of for general-purpose safeguarding, is related to learned optimizers.
>
> Summarizing 1.a and 1.b, we respectfully disagree that the above comments shall be counted as weakness against our work.
>
> More detailed explanation:
> It is precisely because there are numerous papers on implementing algorithms in a deep architecture that this paper is motivated, which we see as a strength. Further, no general purpose safeguarding has been discussed in the learning-to-optimize context. We make the important contribution to establish a general convergence and robustness theory for convex learning-to-optimize, which clearly distinguishes us from previous empirical work on algorithm-driven deep architectures, and also from existing safeguarding algorithms in classical optimization.
>
> To answer your curiosity, we present further analysis on why our theorems are superior to those of the two mentioned papers (even the latter not being relevant to learning-to-optimize):
>
> The SuperMann paper did include a form of safeguarding. Yet, the safeguarding condition in 3(a) of Algorithm 1 of that paper is  i) (first and foremost) not related to learning-to-optimize; and ii) not as general since it compares a candidate iterate only with the current iterate, requiring the safeguarded iterates to be monotone (plus a summable sequence) in the sense of fixed-point residual. Many recent L2O methods are very fast because they are not bound to monotone sequences, so the SuperMann safeguard will slow down those methods. It appears that entire SuperMann Algorithm 1 may be viewed as a special case of our Method 1, depending upon the choice of \mu_k. In addition, our presentation is simpler.
>
> We believe our inclusion of Table 2 is also of practical importance. To make clear distinctions about the safeguarding procedure in this work and its uniqueness, we will add a paragraph on page 2 in the Related Works section summarizing relevant safeguarding papers.
>
> 2) To be brief:
> This is a THEORY paper whose contributions justify themselves mathematically. Previous classical works in the field commonly and sufficiently demonstrate their theories using synthetic experiments, such as “ALISTA: Analytic Weights Are As Good As Learned Weights in LISTA”, ICLR 2019.
>
> More detailed explanation:
> We agree that providing experiments on real-world data (rather than synthetic) would provide a wonderful illustration. However, we disagree with the statement that synthetic results are insufficient to obtain our goal. To clarify, our work set out to identify a framework in which many L2O schemes may be incorporated into to provide theoretical guarantees of their behavior. We did not present any special L2O scheme to compete with state-of-the-art learn to optimize works. Indeed, our first two examples use the L2O schemes from existing works (one of which was published in ICLR) to illustrate how they can be incorporated and what their resulting behavioral differences are.
>
> In situations like MRI (as your reference [3]), being able to ensure that the network does not diverge drastically would be quite important in a clinical setting (which appears to be possible given the literature on fooling neural networks). It would also be bad to have a patient’s MRI come out with artifacts from the network that were unfamiliar to a doctor and subsequently misinterpreted as something malignant. However, in [3] no theoretical guarantees are provided, which implies that such situations are possible. In contrast, if the Deep ADMM-Net were used within our framework, then the outlier cases could potentially be prevented or, at the least, identified by having a flag output from the network if too many safeguarding activations occur.
>
> We respectfully argue that this paper’s main value is not discounted, even though there were few “real data” experiments.
>
> 3) We did make some typos. Those will be updated appropriately.

---

### Official Review · AnonReviewer1 · 2019-10-24
**Official Blind Review #1**

**Rating:** 3

**Review:**

This paper presents a unified framework for parametrizing provably convergent algorithms and learning the parameters for a training dataset of problem instances of interest. The learned algorithm can then be used on unseen problems. One key idea to this algorithm is that it is safeguarded, meaning it will perform some standard, non-learned iterations, if the predicted iterate is not good enough under some condition.

There are three main features of the proposed approach:
1- It unifies various previous approaches such as LISTA, ADMM, non-negative least squares, etc. By defining some operators and safeguarding rules, the same learning approach can be leveraged for these different optimization problems.
2- It is shown that the learned algorithms are provably convergent under some mild assumptions.
3- Empirically, it is shown that the learned algorithms converge faster than the non-learned counterpart on sparse coding, ADMM and non-negative least squares; they use safeguarding sparingly, particularly when used to solve test instances from the same distribution as the training instances.

Additionally, the paper is very well-written. I did not verify the proofs in detail but they seemed OK at a high-level; however, I am not an expert in convex optimization so I hope other reviewers will be able to comment on this aspect.

I do have some deep concerns about the evaluation metrics used to report the results that I will discuss next; these are the main reason for my current score, but I am willing to adjust it if the authors address them convincingly. I also have some comments about related work.

Experimental evaluation:
- The error metric (15) is not suitable for evaluating the performance of an optimization algorithm. You should compute the expectation of the relative error, i.e. E_{d~D} [(f_d(x)-f_d^*) / f_d^*]. This is similar to the average approximation ratio used in the learning to optimize papers for discrete problems (see refs. below). (15) is just the ratio of the expected absolute error to the expected optimal value; I don't think that is equivalent to what I suggested.
- The relative error values are massive in some cases, e.g. Fig. 3. What's going on there? Are all methods performing that horribly? Am I misinterpreting the metric?
- Why do the plots for the seen distribution extend over thousands of iterations but only for tens of iterations for the unseen distribution?
- Please use the same scale for the y-axes in Figs. 1-3.

Methodology:
- Your method requires learning per-iteration parameters. The other L2O methods for gradient descent (see refs. below) use shared parameters instead. This allows them to run for many iterations, possibly beyond what they were trained for. Your method does not allow for that. On the other hand, such models are recurrent and thus possibly more difficult to train than your unrolled feedforward model. Is the fixed number of iterations a limitation of your method? Please discuss this.

Related work:
- Learning for gradient descent: I am surprised these papers are not mentioned although they are quite relevant. They are rather recurrent networks with shared parameters across iterations, but you should also compare against them both conceptually and experimentally:

"Learning to optimize." arXiv preprint arXiv:1606.01885 (2016).
"Learning to learn by gradient descent by gradient descent." Advances in neural information processing systems. 2016.

- Learning to optimize in the discrete setting: there is lots of recent work on this that you should at least point to in passing, e.g.:

"Learning combinatorial optimization algorithms over graphs." Advances in Neural Information Processing Systems. 2017.
"Combinatorial optimization with graph convolutional networks and guided tree search." Advances in Neural Information Processing Systems. 2018.
(Survey) "Machine Learning for Combinatorial Optimization: a Methodological Tour d'Horizon." arXiv preprint arXiv:1811.06128 (2018).

- Theory for learning to optimize: Since you have a theoretical basis for your framework, you should discuss connections to other recent frameworks such as the one below by Balcan et al. It is geared towards the discrete setting and sample complexity rather than convergence, but you should nevertheless discuss it.
Balcan, Maria-Florina, et al. "How much data is sufficient to learn high-performing algorithms?." arXiv preprint arXiv:1908.02894 (2019).

Clarification questions:
- "The choice of parameter ζ k in Line 3 may be any value that results in a well-deﬁned operator T L2O": what is "well-defined" here? that T_{L20} is averaged?

Minor:
- Page 3: "A classic theorem states sequences" -> "A classic theorem states that sequences"
- Appendix proofs: please organize into sections and restate the statements before the proofs.

**Experience Assessment:**

I have published one or two papers in this area.

**Review Assessment: Checking Correctness Of Derivations And Theory:**

I assessed the sensibility of the derivations and theory.

**Review Assessment: Checking Correctness Of Experiments:**

I carefully checked the experiments.

**Review Assessment: Thoroughness In Paper Reading:**

I read the paper at least twice and used my best judgement in assessing the paper.

---

> ### Author Response · Authors · 2019-11-10
> **Response to Reviewer 1**
>
> We appreciate your thoughtful and thorough remarks, and for appreciating our work’s merits. Each concern you listed is clarified in our revised submission and point-by-point responses are provided below. We believe our responses below address your concerns and hope, upon reading these, you will increase your score.
>
> First we address the experimental evaluation remarks.
>
> 1) We have updated the plots to instead use the difference in expected error, i.e., \E[ f_d(x^K) - f*_d ].  In the references you listed, we found various values used for the y-axis (including loss value in several cases). In addition to resolving the identified issue, this should be easier to interpret than a relative error (since no extra explanation is needed and it is clear 0 is the optimal plot value). You were correct that we blundered by inaccurately using relative error, and we are happy to fix this. Thanks for pointing that out.
>
> 2) You are correct that the “relative error” that we obtained was large, particularly in Figure 3. This is simply because several thousand iterations are required to obtain convergence by the KM method and we limited the example layers to the accuracy of about one thousand iterations of the KM method.
>
> 3) The reason for the different number of iterations shown in the plots for “seen” versus “unseen” distributions is that the emphasis of the plots is quite different. In the case of the seen distribution, the goal is to illustrate how the LSKM method compares to the reference KM method (point (i) at the top of page 7). However, in the case of the unseen distribution, the goal is to show the different behaviors of the safeguarded and unsafeguarded methods (point (ii) at the top of page 7), as they will clearly start to diverge in the very early iterations, To humor curious readers about overall convergence in the unseen case, we have updated the x-axis of plots to be log-scale and increased the number of iterations shown.
>
> 4) Please note that it would not be comparing apples to apples if one were to fix the y-axis scale and compare between plots. The objective functions are defined in terms of the data d. Thus, even for the same experiment (e.g., ALISTA in Figure 1), because the underlying distributions in 1a and 1b are different, the resulting average objective function values will be different. We believe it is only meaningful to compare curves within the same plot. The second paragraph in Section 5 has been revised accordingly.
>
> Methodology:
> Our method does NOT  “require” learning per-iteration parameters. In our experiments we chose to use layer-dependent weights since we believe this yields superior performance for a fixed number of iterations in our experiments. To clarify this matter, we have revised the wording in Section 4 to include a set C specifying the network structure and added Remark 4.2 on page 6 where we note one could use layer-independent weights. Such a situation would still be covered by Theorem 3.1 since the theorem’s claim is dependent only on T and \mu_k, not T_{L2O}.
>
> Related work:
> Thank you for pointing out these related works. We initially focused our references on those who learn solve convex optimization problems. We agree that we should have included a more comprehensive discussion of the general learn-to-optimize literature, and appreciate the list of papers that you kindly point out. Following your suggestion, we have added further discussion in the “Related Works” section at the beginning of the paper. We plan to add more detailed discussions of those related work after we revise the paper further to save more space.
>
> Clarifying questions:
> We mean well-defined in the usual sense where a function f(x) is well-defined if to each input x there is a unique identifiable output f(x). So, taking f(x) = T_{L2O}(x, \zeta),  we assume f(x) is well-defined. Also, each averaged operator is nonexpansive, and so this was indirectly addressed in the same sentence of our submission as where your question is drawn from. However, your comment reveals our lack of clarity, which we have now resolved by replacing “nonexpansive” with “averaged” on line 5 of page 4. We have also removed each instance of “firmly nonexpansive” from the paper and used “averaged” where appropriate to further make things clear.
>
> Minor:
> - Fixed.
> - Noted. This requested change has been made.
>
> Additional Revisions:
> - We updated Section 5.3 since there were errors in (24) and the following paragraph of our initial submission.
>
> - Table 1 was revised to add 3 methods.
>
> - The first paragraph in Section 6 has a few minor revisions to reduce length (so that our paper complies with the space limit).
>
> - Figure 3 has been moved to the Appendix to comply with the space limit.
>
> - The word “Because” was removed from the first paragraph in the proof of Theorem 3.1.

---

> > ### Comment · AnonReviewer1 · 2019-11-13
> > **Rebuttal feedback**
> >
> > Thanks for the update folks. I can see some improvements and you've addressed some of my concerns. Some comments:
> >
> > - As I mentioned in my review, the best would be to use the mean relative error: E_{d~D} [(f_d(x)-f_d^*) / f_d^*], rather than just the numerator (absolute error) as you've done in the revised paper. Using the mean relative error allows you to fix the y-axis range so that one can interpret performance on seen vs unseen. The mean relative error is easy to interpret in seen vs unseen, whereas mean absolute error is not, as you've correctly explained: "because the underlying distributions in 1a and 1b are different, the resulting average objective function values will be different". The reader should be able to understand how much worse the algorithm is on unseen data compared to what it was trained on.
> >
> > - I agree with Reviewer #2 on the breadth of the experiments: I recommend comparing against some other methods and on more datasets. I don't see a reply to Reviewer #2's review.
> >
> > Both of these comments are trying to get an answer to the following question: is the proposed framework of practical relevance? Or is it a comprehensive theoretical framework of little use?
> >
> > One could argue that the paper's main contribution is theoretical/algorithmic and the experiments serve only to confirm the intuition behind the theory and so they need not be super extensive. However, it seems that you have already implemented everything needed to expand the experimental results to other learning-based algorithms and datasets. That being said, I don't expect to modify my Rating at the moment.

---

> > > ### Author Response · Authors · 2019-11-15
> > > **Response to Reviewer 2 Rebuttal Feedback**
> > >
> > > Thanks again for your feedback. We address each comment in turn.
> > >
> > > 1) We acknowledge your point and will update the plots to use mean relative error, as suggested.
> > >
> > > 2) Please see our recently posted response to Reviewer 2. In short, "This is a THEORY paper whose contributions justify themselves mathematically."
> > >
> > > 3) We believe the proposed framework is of practical relevance, particularly for its strong guarantee that extreme outliers will not occur (again see our response to Reviewer 2). It is NOT our aim to improve the performance of an L2O method in the average case. Rather, we seek to allow the L2O method to perform in its desired manner, except for situations where it diverges (e.g., as illustrated in Figure 1b).
> > >
> > > 4) Indeed, our main contribution is algorithmic and our experiments do support our theoretical intuitions. Your comment " However, it seems that you have already implemented everything needed to expand the experimental results..." should be considered praise rather than a shortcoming. Indeed, it is our aim to make these algorithmic results easily accessible and able to be applied by practitioners.

---

### Official Review · AnonReviewer4 · 2019-11-01
**Official Blind Review #4**

**Rating:** 3

**Review:**

This paper is trying to provide a general learning-to-optimize(L2O) convergence theory. It proposes a general framework,  the Learned Safeguarded KM(LSKM) method, and proves the convergence of the algorithms generated by this method under certain conditions. Both the theoretical results and the experimental findings have been presented.

This paper should be rejected because it does not properly answer the problem it is trying to address. (1) The LSKM method with any \mu_k is the universal method and it encompasses all L2O algorithms when the safeguarding condition \|S(y^k)\|<= (1-\delta) \mu_k always holds.  However Assumption~3 cannot cover the cases that the safeguarding condition always holds. Thus Theorem 3.1 gives the convergence of some algorithms generated by the LSKM method rather than the convergence of L2O schemes. (2) Theorem~3.1 is only related to the safeguarding procedure and the convergence of T. If we replace T_{L2O} by other operators, Theorem~3.1 still holds.  In my view, this work provides a practical technique to guarantee the convergence of L2O algorithms rather than a general L2O convergence theory.

Also I have some comments as follow:
1. Section~2 provides an overview of the fixed point method. However only a few definitions and notations in this section is helpful to understand the proposed method. Please shorten this part.
2. Dose the safeguarding procedure guarantee the convergence of the LSKM method and decrease the convergence rate comparing to the corresponding L2O algorithm? Please explain more about the role of the safeguarding procedure.
3. It would be better to have a real data example in Section~5.



**Experience Assessment:**

I do not know much about this area.

**Review Assessment: Checking Correctness Of Derivations And Theory:**

I assessed the sensibility of the derivations and theory.

**Review Assessment: Checking Correctness Of Experiments:**

I assessed the sensibility of the experiments.

**Review Assessment: Thoroughness In Paper Reading:**

I read the paper at least twice and used my best judgement in assessing the paper.

---

> ### Author Response · Authors · 2019-11-12
> **Response to Reviewer 4**
>
> Thank you for your careful review and comments. To our best understanding, a few comments seem to arise from misinterpreting our paper’s content: we apologize if our manuscript has not been more clear and might have caused those confusions. Respectfully, we feel we have to disagree with the statement that “this paper should be rejected because it does not properly answer the problem it is trying to address”.
>
> We reply to your remarks below in the order that they were given. We sincerely hope they will clarify your concerns and convince you to increase the score.
>
> Your comment: “(1) The LSKM method with any \mu_k is the universal method and it encompasses all L2O algorithms when the safeguarding condition \|S(y^k)\|<= (1-\delta) \mu_k always holds.  However Assumption~3 cannot cover the cases that the safeguarding condition always holds.”
>
> Answer: We respectfully disagree. The case “the safeguarding condition always holds” is covered by the second half of Assumption 3, quote “the sequence {µk}_k∈N converges to zero.” In other words, Assumption 3 allows the inequality in Line 6 to hold either finitely many or infinitely many times, and in the latter case (which is your concern), the sequence {µk}_k∈N must converge to zero. In addition, we prove this convergence does hold for all the choices of µk listed in Table 2. To see this, in Corollary 3.1, we state “... {µk}_k∈N is generated using a scheme outlined in Table 2, then Assumption 3 holds …”
>
> To resolve any possible ambiguity, we revised the sentence of Assumption 3 to now read “If the inequality in Line 6 holds infinitely many times, then {µk}_k∈N converges to zero.”  This revision should make clear that all possible cases of safeguard conditions holding/failing are covered by our result.
>
> Your comment: “(2) Theorem~3.1 is only related to the safeguarding procedure and the convergence of T. If we replace T_{L2O} by other operators, Theorem~3.1 still holds.  In my view, this work provides a practical technique to guarantee the convergence of L2O algorithms rather than a general L2O convergence theory.”
>
> Answer: True, but in our opinion, it is a blessing, not a curse. To see this, it is important to notice that, while all L2O methods aim for fast optimization, their operators T_{L2O} are very diverse, having many different forms and properties. A safeguard, therefore, must be robust. It is precisely our intention to make Theorem 3.1 to fit any operator. Thus, the dependence of Theorem 3.1 only upon T and \mu_k reveals its generality rather than its limitations.
>
> Theorem 3.1 is especially suitable for L2O operators, whose iterates are often non-monotonic. Therefore, our safeguard was especially designed to be robust to this behavior. When L2O works well on a problem, the safeguard will not intervene unnecessarily.
>
> We agree with you that this work provides a practical technique/framework for guaranteeing convergence of L2O algorithms. And, an equally general result about general L2O convergence (without safeguarding) may not possibly exist since any such result would be highly dependent upon the distribution of data and be necessarily prescribed in a probabilistic manner (thus unable to well-handle outliers, which may be of utmost importance in medical applications). Thus, we wish to reemphasize the point that safeguarding gives a certain guarantee so one can safely apply L2O to data that are possibly unseen.
>
> Below is an itemized response to your other comments:
>
> - Noted. We have removed all material related to firm nonexpansiveness as the essential property we use later in the paper is averagedness, which is more general.
>
> - The corresponding L2O algorithm may not necessarily have any theoretical guarantees of convergence, let alone a convergence rate.
>
> -This is a fair point. However, we believe our synthetic data gives greater control over the “seen” and “unseen” distributions. Please see 2) in our response to Reviewer 2 below (that will be posted soon).

---

> > ### Comment · AnonReviewer4 · 2019-11-14
> > **Rebuttal feedback**
> >
> > Thank you for your point-to-point answer. I have some comments and questions as follow:
> >
> > Re:  Your answer to Comment (1): We respectfully disagree. The case “the safeguarding condition always holds” is covered by the second half of Assumption 3, quote “the sequence {µk}_k∈N converges to zero.” In other words, Assumption 3 allows the inequality in Line 6 to hold either finitely many or infinitely many times, and in the latter case (which is your concern), the sequence {µk}_k∈N must converge to zero. In addition, we prove this convergence does hold for all the choices of µk listed in Table 2. To see this, in Corollary 3.1, we state “... {µk}_k∈N is generated using a scheme outlined in Table 2, then Assumption 3 holds …”
> >
> > 1. Theorem~3.1 can prove the convergence of the LSKM method with the sequence $\{\mu_k\}$ converges to zero. However the LSKM method with $\mu_k\rightarrow 0$ as $k\rightarrow \infty$ cannot encompass all L2O algorithms, and the LSKM method with $\mu_k = \infty$, $k=1,2,...$ encompasses all L2O algorithms.
> >
> > 2. If a L2O algorithm is convergent, does the T_{L2O} operator satisfy the safeguarding condition $\|S(y^k)\| \leq (1-\delta) \mu_k$, $\mu_k \rightarrow 0$ infinitely many times? Is there a convergent L2O algorithm satisfies the safeguarding condition $\|S(y^k)\| \leq (1-\delta) \mu_k$, $\mu_k \rightarrow 0$ finitely many times?
> >
> > Re: Your answer to Comment (2): True, but in our opinion, it is a blessing, not a curse. To see this, it is important to notice that, while all L2O methods aim for fast optimization, their operators T_{L2O} are very diverse, having many different forms and properties. A safeguard, therefore, must be robust. It is precisely our intention to make Theorem 3.1 to fit any operator. Thus, the dependence of Theorem 3.1 only upon T and \mu_k reveals its generality rather than its limitations.
> >
> > 3. Is Theorem~3.1 a general convergence theory of L2O schemes or a general convergence theory of the safeguarding procedures? Please point out the generality of Theorem~3.1.
> >
> > Re: The corresponding L2O algorithm may not necessarily have any theoretical guarantees of convergence, let alone a convergence rate.
> >
> > 4. In Section~5.2, dose the L2O scheme perform better than the LSKM method?
> >
> > 5. There is a 'No guard' method in Table~3, which is the LSKM method without safeguarding. What is the difference between the 'No guard' method and the L2O scheme?

---

> > > ### Author Response · Authors · 2019-11-15
> > > **Rebuttal Feedback Response**
> > >
> > > Thank you for your timely reply. We again believe the main discrepancies arise from fundamental misinterpretations our paper. We respond to each of your 5 points in the number they were given.
> > >
> > > 1. We disagree. First of all, just to clarify, an L2O operator does not own $\mu_k$. The sequence $\{mu_k\}$ is a part of our Algorithm 2, LSKM, which “wraps around” a given L2O operator. Secondly, we reassure that Theorem 3.1 itself is technically correct.
> > >
> > > Now, to address your question, Theorem 3.1 proves convergence under Assumptions 1-3, not under the assumption $\mu_k\rightarrow 0$. Theorem 3.1 covers the case: $\mu_k$ does NOT converge to 0. To see this, notice that, by the “if … then ... else ...” on Line 6 of Algorithm 1 and Assumption 3, the case that “mu_k does NOT converge to 0” can occur ONLY when the L2O operator is applied finitely many times, hence the classic operator on Line 9 of Algorithm 1 (which comes with convergence guarantees) is applied infinitely many times and makes $\{x^k\}$ converge.
> > >
> > > For example, consider the identity operator as a dummy L2O operator. This is a bad L2O operator. It will fail the “if” condition Line 6, thus causing Line 9 (the classic operator) to run infinitely many times and converge.
> > >
> > > Therefore, Theorem 3.1 covers the case in question.
> > >
> > > 2. To your questions, no claim can be made here in general. The number of times the safeguarding would be activated depends on the parameters chosen for $\mu_k$. In particular, even if the L2O algorithm converges by itself, the safeguarding condition still fails either finitely many or infinitely many times, since the condition asks for sufficient progress (a point realized by the choices in Table 1). After all, convergence is an asymptotic concept, and a convergence rate can be arbitrarily slow.
> > >
> > > 3) Safeguarding is used to guarantee convergence with any input L2O algorithm. It is a general convergence theory of the safeguarding procedures.  This may be used for any L2O algorithm, including those created solely through heuristics.
> > >
> > > 4) Please see Figure 1b. With an appropriate safeguarding choice, LSKM is equally good or better than an unsafeguarded L2O.
> > >
> > > 5) “No guard” is used to mean that $\mu_k = \infty$ is used the in the LSKM algorithm, i.e., the original $T_{L2O}$ update operation is always used at each step.

---

### Decision · Program_Chairs · 2019-12-19

**Decision:**

Reject

**Comment:**

This paper gave a general L2O convergence theory called Learned Safeguarded KM (LSKM).  The reviewers found flaws both in theory and in experiments.  While all the reviewers have read the authors' rebuttal and gave detailed replies, they all agree to reject this paper.  I agree also.